# Pressure-driven release of viral genome into a host nucleus is a mechanism leading to herpes infection

Alberto Brandariz-Nuñez[1], Ting Liu[2], Te Du[3], Alex Evilevitch[1,4]*

[1]Department of Pathobiology, College of Veterinary Medicine, University of Illinois at Urbana-Champaign, Urbana, United States; [2]Department of Physics, Carnegie Mellon University, Pittsburgh, United States; [3]The Marjorie B. Kovler Viral Oncology Laboratories, The University of Chicago, Chicago, United States; [4]Department of Experimental Medical Sciences, Lund University, Lund, Sweden

**Abstract** Many viruses previously have been shown to have pressurized genomes inside their viral protein shell, termed the capsid. This pressure results from the tight confinement of negatively charged viral nucleic acids inside the capsid. However, the relevance of capsid pressure to viral infection has not been demonstrated. In this work, we show that the internal DNA pressure of tens of atmospheres inside a herpesvirus capsid powers ejection of the viral genome into a host cell nucleus. To our knowledge, this provides the first demonstration of a pressure-dependent mechanism of viral genome penetration into a host nucleus, leading to infection of eukaryotic cells.
DOI: https://doi.org/10.7554/eLife.47212.001

## Introduction

Recent studies have found that many families of viruses have highly stressed packaged genomes, exerting tens of atmospheres of pressure, inside their viral protein shell, termed the capsid [e.g. bacteriophages (*Evilevitch et al., 2003*), archaeoviruses (*Hanhijärvi et al., 2013*) and eukaryotic viruses (*Bauer et al., 2013*), infecting all three domains of life]. The pressure results from tight confinement of the negatively charged double-stranded (ds) viral DNA or dsRNA inside the capsid (*Tzlil et al., 2003*; *Kindt et al., 2001*; *Purohit et al., 2005*). Our recent measurement of 20 atmospheres of DNA pressure in a Herpes simplex type 1 (HSV-1) capsid (*Bauer et al., 2013*) was the first demonstration of a pressurized genome state in a eukaryotic virus. This high internal capsid pressure is generated by an ATP-driven packaging motor located at a unique capsid vertex, shown to be the strongest molecular motor known (*McElwee et al., 2018*; *Smith et al., 2001*). Structural features of packaging motor components are shared by bacterial and archaeal dsDNA viruses and eukaryotic herpesviruses (*Krupovic and Bamford, 2011*). This strongly suggests that once DNA is packaged with high force into a capsid, the reverse process of pressure-driven genome release is one of the central mechanisms of viral replication. A previous attempt to demonstrate this mechanism analyzed the velocity of DNA ejection from phage λ into an *E. coli* cell to determine whether ejection dynamics correlates with a decrease in intracapsid DNA pressure (*Van Valen et al., 2012*). However, due to large cell-to-cell variability in the ejection rates, the results were difficult to interpret. [We recently found, using isothermal titration calorimetry, similar timescale variability in phage λ ejection dynamics, ranging from a few seconds to minutes; this variability is caused by the metastable state of the tightly packaged genome resulting from DNA-DNA electrostatic sliding friction, which can delay or stall the ejection process despite high pressure in the capsid. However, this interstrand friction was significantly reduced by a transition in intracapsid DNA structure induced by optimum environmental conditions favorable for infection, leading to essentially instant DNA

*For correspondence:
alexe@illinois.edu

Competing interests: The authors declare that no competing interests exist.

release (*Evilevitch, 2018*).] Thus, the role that high DNA pressure in phage capsids might play in viral genome delivery into a bacterial cell remained unclear. Furthermore, the experimental evidence placing the discovery of intracapsid genome pressure (*Bauer et al., 2013*) in the context of eukaryotic viral infection was lacking. Here, we conducted a stringent test, showing that DNA pressure in HSV-1 capsids powers ejection of the viral genome into a cell nucleus. This provides, to our knowledge, the first demonstration of a pressure-dependent mechanism leading to infection of eukaryotic cells (where the term 'infection' denotes the introduction of viral nucleic acid into a host cell by a virus [*Flint, 2004*]).

*Herpesviridae* are a leading cause of human viral disease, second only to influenza and cold viruses (*Pellett and Roizmann, 2007*; *Roizmann et al., 2007*; *Sandri-Goldin, 2006*). The herpesviridae family includes a diverse set of viruses, nine of which are human pathogens (*Davison et al., 2009*). Herpesvirus infections are life-long with latency periods between recurrent reactivations, making treatment difficult (*Davison et al., 2009*; *Pai and Weinberger, 2017*). Herpesvirus infections frequently reactivate to result in recurrent acute oral and genital lesions, encephalitis (*Roizmann et al., 2007*), shingles (*Coen, 2006*), birth defects and transplant failures, as well as oncogenic transformation (*Rickinson and Kieff, 2007*; *Ganem, 2007*). Herpesviruses consist of a double-stranded (ds) DNA genome packaged within an icosahedral capsid that is surrounded by an unstructured protein layer, the tegument, and a lipid envelope. *Figure 1* illustrates the HSV-1 infection process as observed by ultrathin-sectioning transmission electron microscopy (TEM). After binding at the outer membrane (*Figure 1a*), viruses enter the cell cytoplasm and are transported toward the nucleus (*Figure 1b*). The viral capsid ejects its genome upon docking to a nuclear pore complex (NPC), which forms a passageway for molecular traffic into the nucleus (*Figure 1c*) (*Sodeik et al., 1997*). To investigate the specific event of herpes DNA injection into a cell nucleus, we designed an assay built on previous experiments showing that purified HSV-1 capsids bind to NPCs on isolated nuclei and eject their DNA into nuclei in the presence of cytosol supplemented with an ATP-regeneration system (*Ojala et al., 2000*). This reconstituted nuclei system allows us to determine if viral DNA is ejected into the nucleus when the capsid pressure is 'turned off' by addition of an external osmolyte. We had shown previously that DNA ejection from isolated HSV-1 capsids into solution can be suppressed by creating an osmotic pressure in the host solution, which matches the pressure of the packaged DNA (*Bauer et al., 2013*). This effectively eliminates genome pressure in the capsid (the mechanism of osmotic suppression is explained below). In this work we show that viral genome ejection through the NPCs into a cell nucleus can be completely suppressed in the presence of the biologically inert osmolyte polyethylene glycol (PEG). The reconstituted nuclei system accurately reproduces capsids-nuclei binding and nuclear transport of the herpes genome into living cells (*Ojala et al., 2000*; *Adam et al., 1990*; *Au et al., 2016*; *Cassany and Gerace, 2008*). It provides the benefit of isolating the effect of eliminated capsid pressure on the single step of viral DNA ejection, while avoiding interference from hyperosmotic conditions on other processes occurring within the cell during viral replication.

To provide evidence that intracapsid DNA pressure is responsible for DNA release from a herpesvirus capsid into a cell nucleus, it is essential to show that when the capsid pressure is 'turned off' with addition of an external osmolyte, herpes capsids bound to NPCs do not eject DNA into a nucleus, while the ejection is completed successfully without osmolyte addition (see illustration in *Figure 1d*). While the term 'infection' usually refers to both viral genome transport into the cell and subsequent replication of the virus, the primary infection by several types of herpesviruses (including HSV-1) is latent (i.e. the herpes genome is translocated into the host nucleus, without subsequent genome replication [*Steiner, 1996*]). Thus, the osmotic suppression assay, combined with the reconstituted nucleus system in this work, present a platform for analysis of a pressure-dependent mechanism of herpesvirus infection focused on the viral genome translocation step.

This paper is divided into three sections. In *Section 1*, we determine the critical PEG concentration at which the DNA pressure in HSV-1 capsids is 'turned off'. In *Section 2*, we validate that the presence of PEG does not affect the integrity and functionality of reconstituted nuclei, as well as the binding of HSV-1 capsids to nuclei. In *Section 3*, we designed a pull-down assay with real-time PCR (qPCR) quantification of the amount of DNA injected into a nucleus when the capsid pressure is 'on' and 'off.' Suppression of DNA injection also is visualized by ultrathin-sectioning electron microscopy (EM).

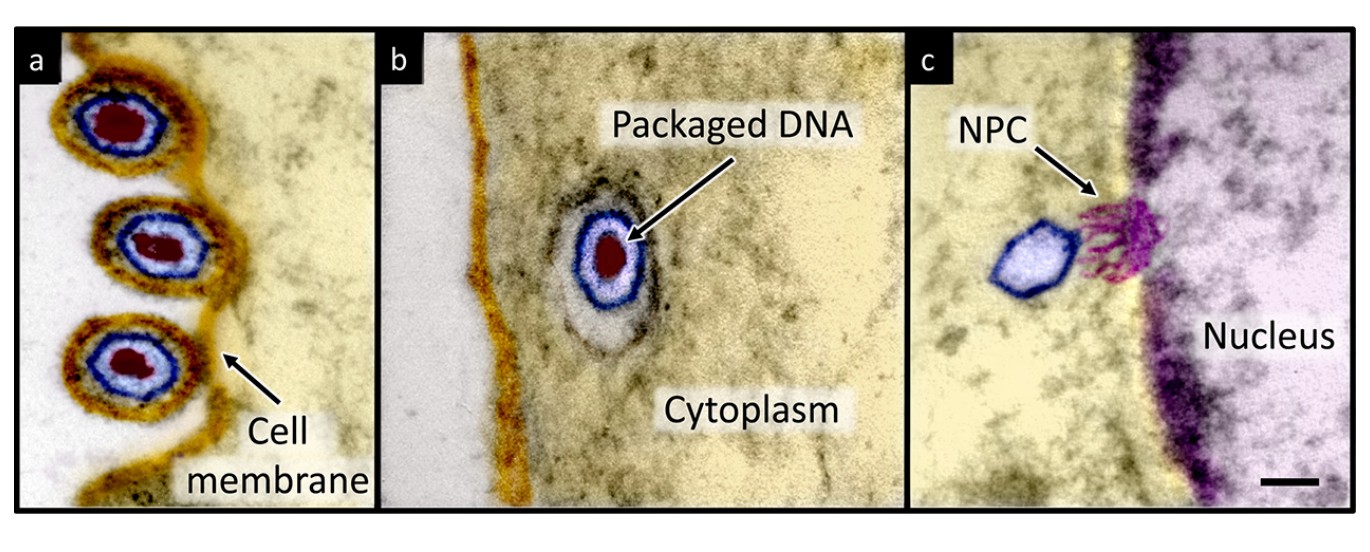

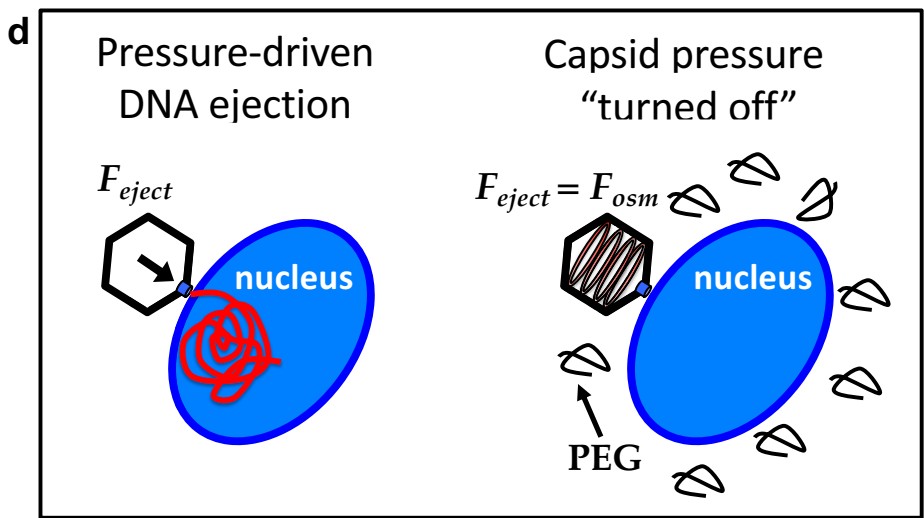

**Figure 1.** Ultrathin-sectioning EM visualization of the HSV-1 infection process showing viral DNA ejection from HSV-1 capsid into a host nucleus. Ultrathin Epon sections of Vero cells infected with HSV-1 at an MOI of 300 PFU/cell. Artificially colored electron micrographs of HSV-1 at the cell membrane (A), in transport to the nucleus (B), and bound at a nuclear pore complex (NPC) embedded within the nuclear envelope (C). The dsDNA genome appears as an electron-dense region within the capsid, which is visible in (A) and (B), but absent in (C) due to DNA ejection upon NPC binding. Scale bar, 50 nm. Adapted for clarity from our earlier publication (*Bauer et al., 2013*). (D) Illustration of the osmotic suppression experiment. DNA ejection from a virus capsid into a reconstituted host nucleus is completed successfully without osmolyte addition. However, viral DNA ejection is fully suppressed, when the capsid pressure is 'turned off' with an external osmotic pressure, created by PEG, that matches the pressure of the packaged DNA in the capsid.

DOI: https://doi.org/10.7554/eLife.47212.002

## Results and discussion

All herpesviruses have strongly confined dsDNA inside the capsid, which is released into a cell nucleus upon the capsid docking to the NPCs at the nuclear envelope (*Heming et al., 2017*). We used HSV-1, which is a prototypical, experimental model system to study herpesvirus replication due to the ease of growing and purifying large quantities of viral C-capsids (DNA-filled capsids without a lipid envelope and tegument proteins)(*Méndez-Álvarez, 2000*).

## 'Turning off' DNA pressure inside a viral capsid with external osmotic pressure

Below we provide a description of the mechanism through which DNA pressure inside the capsid is 'turned off' with osmolyte addition. Viral capsids are permeable to water and small ions (*Trus et al., 1996*; *Heymann et al., 2003*). In an aqueous buffer solution (without osmolyte addition), due to the high DNA concentration inside the capsid, water is drawn into the fixed volume through the capsid pores, as a result of the entropic drive to maximize mixing, and a large osmotic pressure is developed inside the capsid to equalize the chemical potential of water inside and outside of the capsid. This pressure, due to the compressed water inside the rigid capsid volume can also be described as a DNA repulsion and bending pressure withstood by the capsid walls (*Evilevitch et al., 2008*). Addition of an osmolyte to the solution surrounding the capsid, where osmolyte is larger than the capsid pores, creates an osmotic gradient. Water will be drawn out of the DNA volume to dilute the osmolyte molecules outside the capsid. This reduces the water density and the osmotic pressure inside the capsid. Once the osmolyte concentration reaches $c*$, the pressure inside the capsid is brought down to one atm (atmospheric pressure). For this special value of $c*$, the water-exchange equilibrium corresponds to zero osmotic pressure difference inside and outside the capsid and a net force of zero on the interior of the capsid's rigid walls confining the DNA. Thus, the osmotic pressure associated with the osmolyte concentration, $c*$, is equal to the osmotic pressure exerted by the confined DNA. As a consequence, even if the DNA was allowed the opportunity to ''escape'' from its confinement (when the virus capsid is opened), it would not, because there is no driving force for this process. For any lesser value of external osmotic pressure than that provided by $c*$, there is a pressure difference and hence a net force (outward) on the confining capsid walls because an insufficient amount of water has been drawn out of the DNA solution to lower its osmotic pressure to one atm. This explains our previously designed experiment of osmotic suppression of viral DNA ejection where we measured the pressure in HSV-1 capsids (*Bauer et al., 2013*).

First, without nuclei present and using our osmotic suppression assay (*Evilevitch et al., 2003*; *Bauer et al., 2013*), we used solutions containing PEG to determine the critical concentration, $c*_{PEG}$, that matches the DNA pressure in an HSV-1 capsid and thus turns it off. To create an osmotic pressure gradient between the interior and exterior of the capsid, we used PEG with molecular weight MW ≈ 8 kDa, which does not permeate the capsid since the HSV-1 capsid pore diameter is ~20 Å, corresponding to ~4 kDa MW cutoff (*Trus et al., 1996*; *Heymann et al., 2003*). DNA ejection from the capsid was triggered by mild trypsin treatment. Trypsin cleaves the portal 'plug' proteins UL6 and UL25 while the rest of the HSV-1 capsid remains intact (*Bauer et al., 2013*; *Newcomb et al., 2007*). The length of DNA remaining in the capsid as a function of increasing PEG 8 kDa concentration was determined with pulse field gel electrophoresis (PFGE) combined with a DNase protection assay, as described in *Bauer et al. (2013)*. *Figure 2* shows that a progressively smaller fraction of DNA was ejected from HSV-1 capsids with increasing external osmotic pressure, where the DNA ejection was completely suppressed at $c*_{PEG}$PEG30% w/w (PFGE data are shown in *Figure 2—figure supplement 1*). This corresponds to ~18 atm of external osmotic pressure equal to the DNA pressure in the capsid, at buffer conditions required for capsid binding to isolated nuclei, set by capsid binding buffer CBB (see Materials and methods Section). It should be noted that ionic conditions in the surrounding buffer also affect DNA pressure in the capsid through cations permeating the capsid and screening the repulsive DNA-DNA interactions (*Evilevitch et al., 2008*). We used this critical PEG concentration ($c*_{PEG}$) to turn off the capsid pressure when HSV-1 C-capsids were incubated with reconstituted nuclei. In the next Section, we demonstrate specific capsid binding to NPCs at the nuclear membrane and confirm that the nuclear integrity as well as capsid-nuclei binding are not affected by the addition of 30% w/w PEG 8 kDa.

## Reconstituted capsid-nuclei system

Purified HSV-1 C-capsids were incubated with nuclei isolated from rat liver cells supplemented with cytosol and ATP-regeneration system. Cytosol contains importin-β required for efficient HSV-1 capsid binding to NPCs (*Ojala et al., 2000*; *Anderson et al., 2014*). The ATP-regeneration system is not required for capsid binding to NPCs, but it is required for opening the capsid's portal leading to DNA ejection (*Anderson et al., 2014*). The ATP-regeneration system contains ATP and GTP (as well as other components [*Ojala et al., 2000*]), both of which are required for maintenance of the Ran-

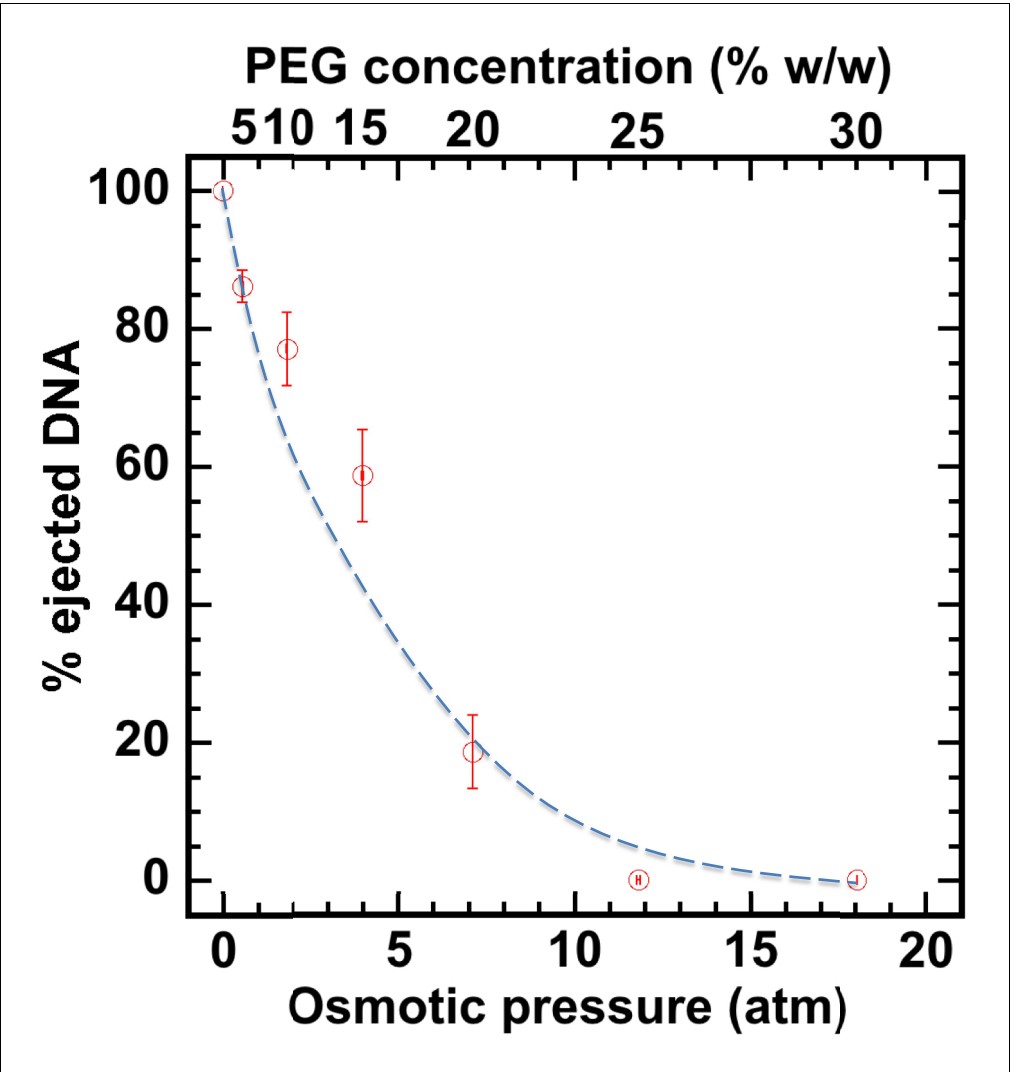

**Figure 2.** Percentage of viral genome ejected from HSV-1 capsids as a function of the external osmotic pressure. DNA ejection from the capsids in vitro is triggered by mild trypsin treatment, which cleaves the portal protein (UL6) without degrading the major capsid protein (VP5) or causing morphological damage to capsids (*Bauer et al., 2013*). Figure shows that DNA ejection is progressively suppressed with increasing PEG concentration at 37°C with PEG 8 kDa. DNA ejection is completely suppressed at 18 atm external osmotic pressure, which matches and therefore 'turns off' the DNA pressure in the capsid. PEG is added to CBB buffer (capsid binding buffer: 20 mM HEPES-KOH with pH of 7.3, 80 mM K-acetate, 2 mM DTT, 1 mM EGTA, 2 mM Mg-acetate, 1 mM PMSF, and 1X CLAP cocktail), required for capsid binding to nuclei. (PEG concentration was converted to osmotic pressure using the relation in ref *Evilevitch et al., 2003*). Vertical error bars represent the standard error of the gel band intensity profile (see Materials and methods Section). Horizontal error bars representing standard error in weighted PEG concentration are negligibly small. Dashed line is drawn to guide the eye.

DOI: https://doi.org/10.7554/eLife.47212.003

The following figure supplement is available for figure 2:

**Figure supplement 1.** Osmotic suppression of HSV-1 genome ejection.

DOI: https://doi.org/10.7554/eLife.47212.004

---

GTP/GDP gradient across the nuclear membrane (*Cole and Hammell, 1998*). While the mechanisms of interactions that mediate capsid docking to NPC and its opening are not clear, importin-β binds to cargo proteins (*Cautain et al., 2015*; *Macara, 2001*), while the Ran-GTP cycle regulates the importin-cargo association (*Cautain et al., 2015*; *Macara, 2001*). A nuclear localization signal (NLS)

and other motifs on viral capsid proteins are involved in this importin binding interaction (*Cole and Hammell, 1998*; *Flatt and Greber, 2015*). Since herpes DNA does not interact with import protein factors, these protein factors are likely involved in the binding and opening of the capsid portal vertex, which triggers genome ejection, but they do not provide the driving force for the actual translocation of the viral DNA across the NPC channel (which is driven by DNA capsid pressure, as we demonstrate below).

*Figure 3* shows individual GFP-labeled HSV-1 C-capsids (green, strain K26GFP, HSV-1 strain expressing GFP-tagged VP26 protein) bound to NPCs on isolated DAPI-stained cell nuclei (blue), imaged with Super-Resolution Structured Illumination Microscopy (SR-SIM) (*Sekine et al., 2017*). SR-SIM provides resolutions down to 120 nm, allowing visualization of individual capsids attached to the nucleus (HSV-1 C-capsid diameter ≈ 125 nm). [We found that tegument-free C-capsids were able to bind efficiently to NPCs and eject DNA into a nucleus. This finding is consistent with a recent study demonstrating that untegumented C-capsids and viral capsids exposing inner tegument proteins on their surface had a similar degree of binding to NPCs (*Ojala et al., 2000*; *Anderson et al., 2014*).] In parallel, using confocal fluorescent microscopy (FM), we confirmed that DNA-filled C-capsids bind specifically to the NPCs (as opposed to random binding to the nuclear membrane). As a control, we used WGA (wheat germ agglutinin) which blocks the NPC [WGA associates with the glycoproteins within the NPC (*Ojala et al., 2000*; *Finlay et al., 1987*) and competes with capsid binding], and prevents capsid binding, demonstrating capsid-NPC binding specificity, see *Figure 3B*. *Figure 3B* also shows that capsid-NPC binding is not inhibited by the addition of 30% w/w PEG 8 kDa.

*Finan et al. (2011)* demonstrates that nuclear transport through the NPCs is not negatively affected by hyperosmotic conditions corresponding to those used in our study (~20 atm). Specifically, the authors reported that, under hyperosmotic stress, the nuclear size decreased while nuclear lacunarity increased, indicating expansion in the pores and channels interdigitating the chromatin. As a result, the rate of nucleocytoplasmic transport increased but only due to the change in nucleus geometry, providing a shorter effective diffusion distance. This sensitivity to hyperosmotic conditions concerned both passive and active transport across the NPCs. At the same time, the authors found that diffusivity within the nucleus was insensitive to the osmotic environment. In agreement with these studies (*Finan et al., 2011*), we observed that, under hyperosmotic conditions (~18 atm at 30% w/w PEG 8 kDa), the nuclei slightly shrunk (*Figure 3—figure supplement 1A*). However, the sub-nuclear structure of heterochromatin DNA was essentially unchanged upon addition of PEG, as visualized by a DAPI stain of nuclear DNA (*Figure 3—figure supplement 1A*, second row). We also confirmed that the integrity of the nuclei was not affected by the addition of 30% w/w PEG 8 kDa by showing that fluorescently labeled 70 kDa dextran is excluded from the nuclei interior with nuclei remaining intact and structured, see *Figure 3—figure supplement 1B*. Finally, we showed that the full transport functionality of NPCs is maintained in the reconstituted nuclei system at an osmotic pressure of ~18 atm generated by PEG. This was verified with a fluorescently labeled NLS (data not shown) (*Miyamoto et al., 2002*). Purified GST-NLS-EGFP recombinant protein, which contains the NLS of the simian virus 40 T antigen fused with glutathione S-transferase (GST) and EGFP fluorescent protein, was used. Purified rat liver nuclei were incubated with cytosolic extracts (as a source of soluble import factors) supplemented with an ATP-regeneration system and a purified GST-NLS-EGFP recombinant protein at ~18 atm external osmotic pressure generated by PEG. This protein was used as a positive import substrate. GST-NLS-EGFP was fully transported into the nucleus through the NPC by an active mechanism, which was detected by fluorescence microscopy (*Miyamoto et al., 2002*; *Tsuji et al., 2007*; *Vázquez-Iglesias et al., 2009*) (see details in the Materials and methods Section).

Together, these findings show that the reconstituted capsid-nuclei system provides a robust assay for investigation of viral DNA ejection into a cell nucleus under hyperosmotic conditions.

## Quantification of intranuclear DNA release from HSV-1 capsids with genome pressure on and off

Once the integrity and functionality of the reconstituted nuclei system at hyperosmotic conditions were verified, we conducted a stringent test demonstrating the role of intracapsid pressure for HSV-1 genome transport into a host nucleus. This was analyzed using a pull-down assay which allows

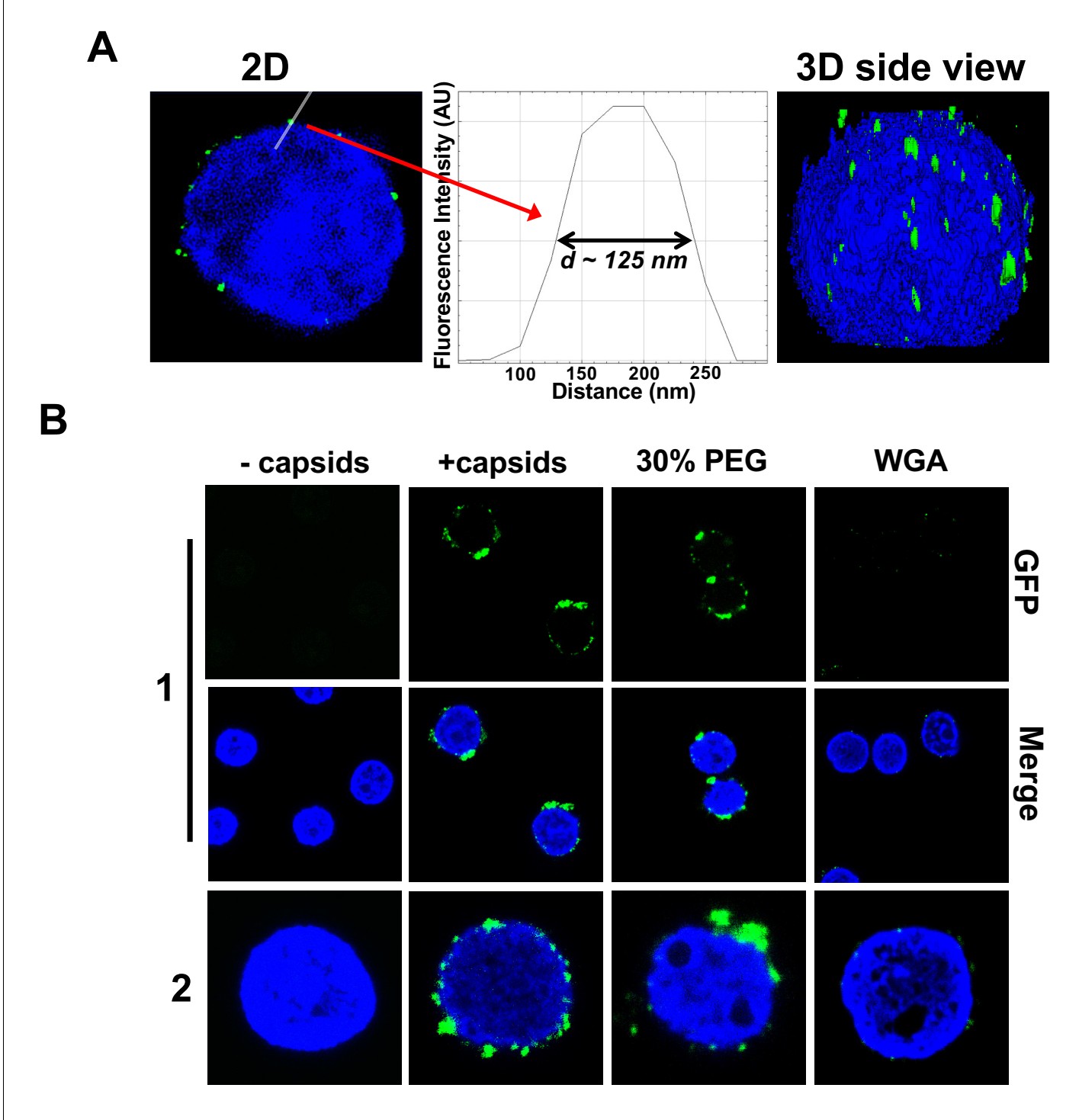

**Figure 3.** Imaging of reconstituted capsid-nuclei system confirms specific capsid binding to the NPCs at the nuclear membrane with and without PEG 8 kDa present. (**A**) Representative super-resolution SIM image showing GFP-HSV-1 C-capsids (green) bound to isolated reconstituted rat liver nuclei (blue DAPI stain). A histogram of a capsid cross-section profile for a capsid GFP signal along the white line shows that individual C-capsids are resolved (HSV-1 C-capsid diameter ≈ 125 nm). (**B**) Confocal fluorescence microscopy images show that binding of GFP-HSV-1 C-capsids (green) to DAPI-stained isolated nuclei (blue), in the presence of cytosol (no ATP-regeneration system was added since it is not required for capsid binding [*Ojala et al., 2000*]), is not inhibited by the addition of 30% w/w PEG 8 kDa. The addition of wheat germ agglutinin (WGA) prevents most of the capsid binding to nuclei, which demonstrates that capsids bind specifically to NPCs as opposed to binding anywhere on the nuclear membrane [WGA associates with the

*Figure 3 continued on next page*

*Figure 3 continued*

specific glycoproteins within the NPC and competes with capsid binding (*Ojala et al., 2000*; *Finlay et al., 1987*)]. The images at the bottom of row two are a zoom-in of the individual nuclei.

DOI: https://doi.org/10.7554/eLife.47212.005

The following figure supplement is available for figure 3:

**Figure supplement 1.** Effect of osmotic pressure on morphology and permeability of nuclei.

DOI: https://doi.org/10.7554/eLife.47212.006

quantification of the amount of DNA injected from HSV-1 capsids into cell nuclei when the capsid pressure is 'on' or 'off', modulated by osmolyte addition.

First, the pull-down assay was used to demonstrate DNA ejection from capsids into nuclei without PEG present (when the capsid is pressurized). Purified HSV-1 C-capsids were incubated with reconstituted nuclei in a CBB buffer at 37°C for 40 min, supplemented with cytosol and ATP-regeneration system. After incubation, capsids bound to nuclei were pelleted and separated from the extranuclear solution by low-speed centrifugation, as illustrated in *Figure 4*. The supernatant with extranuclear solution contains unbound capsids and free viral DNA from broken capsids. The pellet of nuclei with bound capsids was then resuspended in a surfactant-containing buffer to break the nuclear membranes and release into solution the bound capsids and the nucleoplasm contents with injected viral DNA. In the pull-down assay (*Figure 4*), we used anti-HSV-1/2 ICP5/UL19 antibody attached to Protein A beads to immunoprecipitate capsids present in the resuspended nuclear pellet. Analogously, immunoprecipitation was used to separate the unbound viral capsids from the extranuclear solution (the supernatant in *Figure 4*). During all separation and purification steps, the samples were kept at 4°C, which prevents DNA ejection from the capsids after the initial capsid-nuclei 40 min incubation was completed (*Newcomb et al., 2007*). In order to extract and quantify the amount of DNA retained in the capsids, protease K was added to digest the capsid shell. As illustrated in *Figure 4*, by combining this pull-down assay with repeated low-speed centrifugation-separation steps, we successfully separated four fractions of HSV-1 DNA originating from: (a) DNA extracted from capsids that failed to bind to NPCs, (b) free DNA in the extranuclear solution from broken capsids, (c) DNA retained inside the capsids that were bound to nuclei but did not eject DNA, and (d) DNA ejected from capsids into the nucleoplasm, see *Figures 4* and *5A*. Viral DNA in each fraction was further purified using phenol-chloroform extraction (see details in the Materials and methods Section). Combined, these four DNA fractions constitute the total viral DNA load in the capsid-nuclei sample. This pull-down assay allows accurate quantification of the amount of HSV-1 DNA released from capsids bound to NPCs into the nucleoplasm (*fraction d*) (excluding viral DNA from unbound capsids), relative to the total DNA amount in capsids bound to nuclei, which have either ejected or retained their genome (*fractions c+d*).

The amounts of DNA extracted from each fraction (*a,b,c,* and *d*) were quantified by qPCR using specific HSV-1 primers for VP16/UL48 and ICP0 genes. Viral gene copies were compared to PCR amplification of HSV-1 DNA with a known copy number (see Materials and methods Section). Histograms in *Figure 5B* show the DNA copy number for each of the four viral DNA-containing fractions. *Figure 5C* shows the total DNA copy number from *fractions a, b, c,* and *d*. *Figure 5D* shows the fraction of DNA ejected from nuclei-bound capsids (*fraction d/fractions c+d*). After nuclei incubation with HSV-1 C-capsids at 37°C for 40 min without osmolyte addition, *Figure 5D* shows that ~98% of all nuclei bound capsids ejected their DNA into nuclei. All qPCR data for DNA copy numbers in each fraction obtained with the pull-down assay and shown in *Figure 5* are also summarized in a table in *Supplementary file 1*.

Separately, using the fluorescently labeled 70 kDa dextran exclusion assay described in Section two above, we confirmed that viral DNA injection into nuclei did not affect the integrity of the nuclei (*Figure 4—figure supplement 1*). The efficiency of the pull-down assay was assessed by the DNase protection method where we showed that 94–99% of the capsids in a given sample fraction are immunoprecipitated (see method description and data in Materials and methods Section and table in *Supplementary file 1*). The fact that DNA copy numbers determined with the VP16 primer are generally higher than those determined with the ICP0 primer is related to the difference in qPCR amplification efficiency due to differences in primer-gene interactions. This demonstration, showing

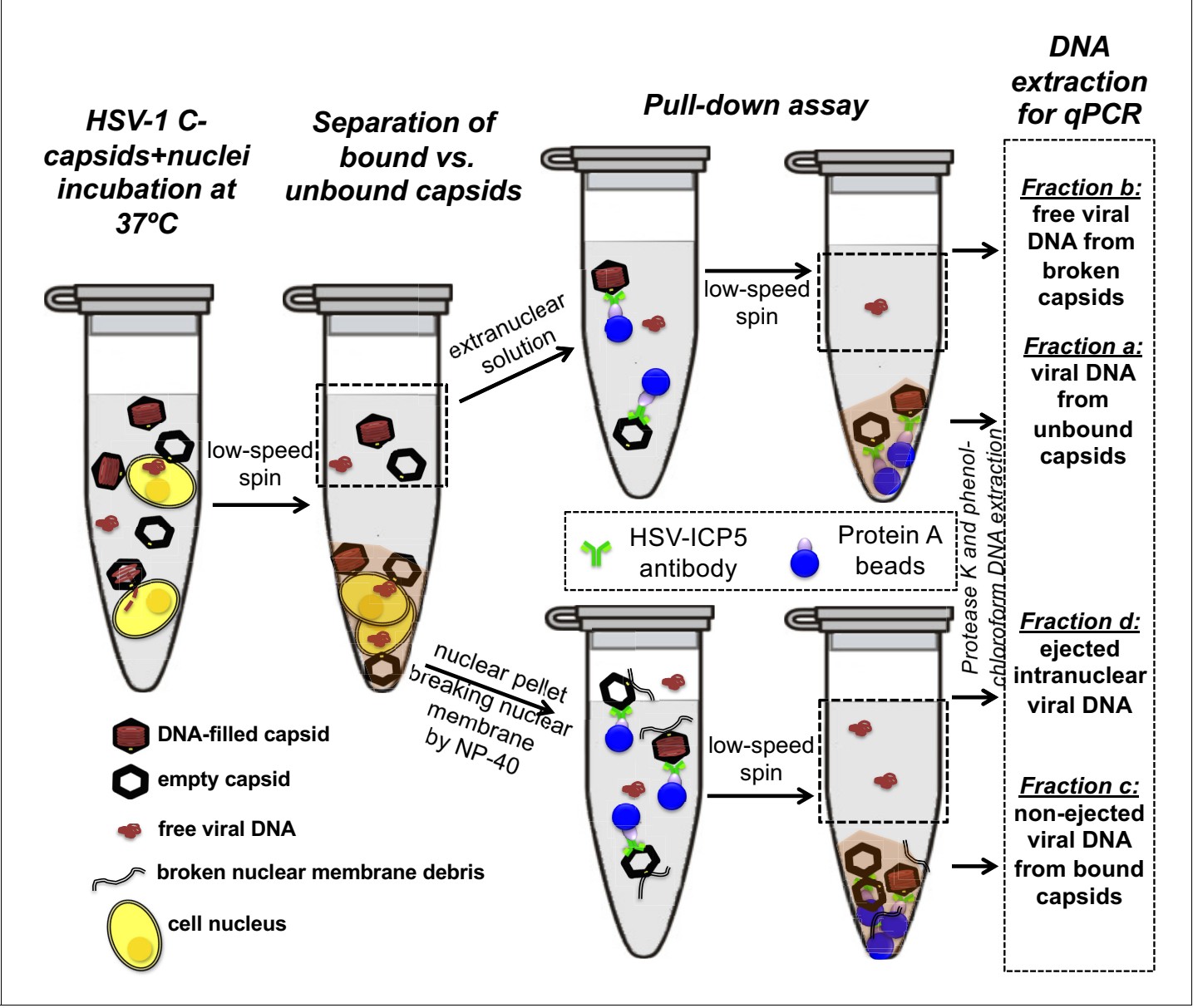

**Figure 4.** Schematic of the pull-down assay for quantification of the amount of DNA injected from HSV-1 capsids into cell nuclei when the capsid pressure is 'on' or 'off', modulated by PEG addition. HSV-1 C-capsids were incubated with reconstituted rat liver cell nuclei in CBB buffer, with and without 30% w/w PEG 8 kDa (osmolyte is not shown in the sketch). This pull-down assay successfully separates four fractions of HSV-1 DNA originating from: (a) DNA extracted from capsids that failed to bind to NPCs, (b) free DNA in the extranuclear solution from broken capsids, (c) DNA retained inside the capsids that were bound to nuclei but did not eject DNA, and (d) DNA ejected from capsids into the nucleoplasm. Viral DNA in each fraction was further purified using phenol-chloroform extraction prior to qPCR quantification. Note that the anti-HSV-1/2 ICP5 antibody attached to Protein A beads for the immunoprecipitation step has multiple binding sites on the capsid but only one antibody bound to a capsid is shown for clarity of presentation. Nuclear chromosomal DNA present in *fraction c* is not shown.

DOI: https://doi.org/10.7554/eLife.47212.007

The following figure supplement is available for figure 4:

**Figure supplement 1.** The integrity of isolated nuclei is not disrupted by DNA ejection from bound HSV-1 capsids.
DOI: https://doi.org/10.7554/eLife.47212.008

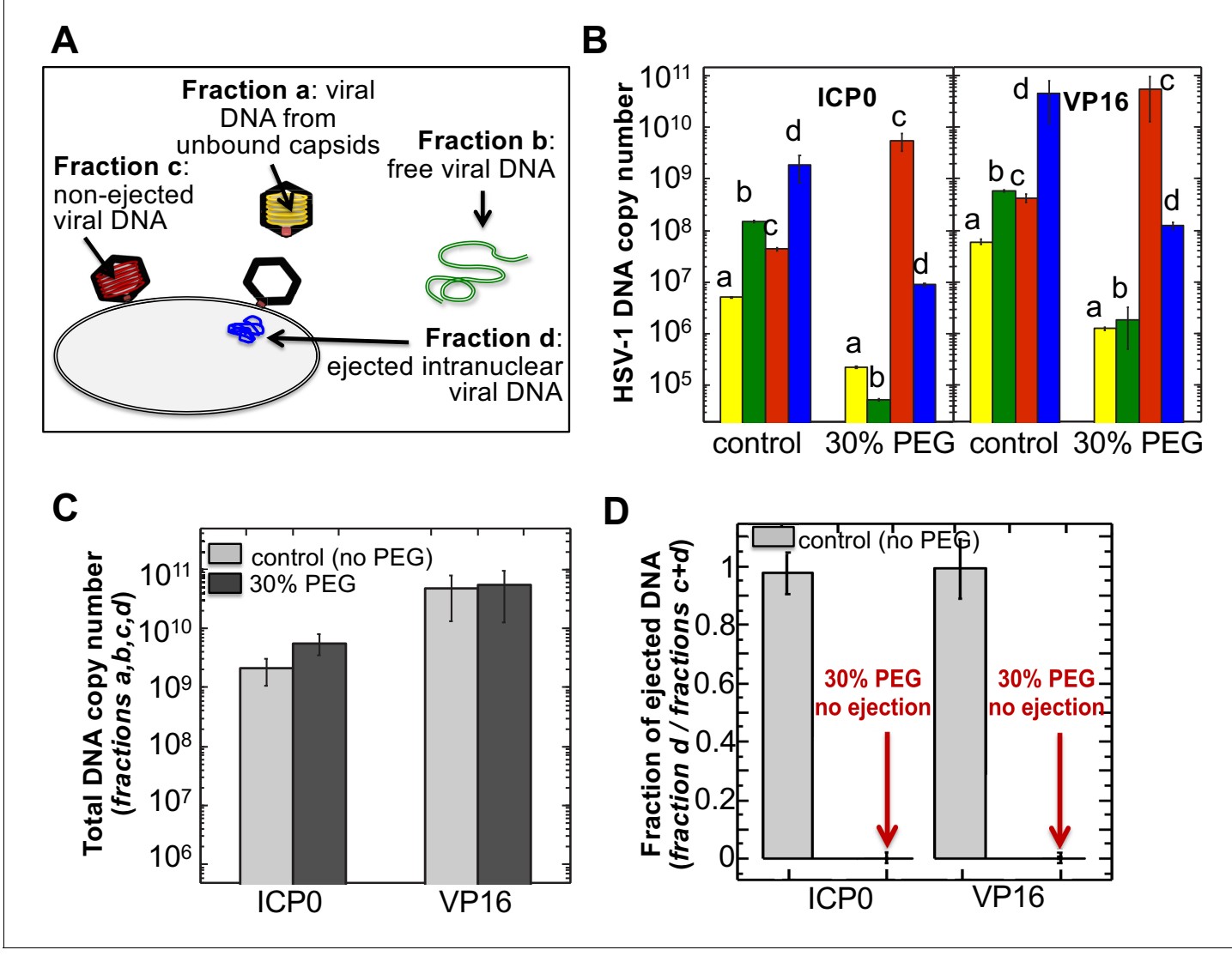

**Figure 5.** The amounts of HSV-1 DNA released into the nucleoplasm from capsids bound to NPCs quantified by qPCR after extraction from each fraction (a,b,c, and d). Specific HSV-1 primers for VP16/UL48 and ICP0 genes were used as amplicons for the qPCR assays of viral DNA copy numbers. (A). Schematic of viral DNA fractions separated from the reconstituted capsid-nuclei system with pull-down assay. (B) Histograms show the DNA copy number for each of the four viral DNA-containing fractions. (C) Histograms show the total DNA copy number from *fractions a, b, c,* and *d* in the reconstituted capsid-nuclei system. (D) Fraction of DNA ejected from nuclei-bound capsids (*fraction d/fractions c+d*). After nuclei incubation with HSV-1 C-capsids at 37°C for 40 min, without osmolyte addition,~98% of all nuclei bound capsids ejected their DNA into nuclei. When the capsid pressure is turned off at 18 atm of external osmotic pressure (generated by addition of 30% w/w PEG 8 kDa), the ejection of DNA from capsids bound to nuclei is completely suppressed (*fraction d/fractions c+d* ~ 0.2%). qPCR DNA copy number quantification is based on a standard curve generated by serial dilution of a wild-type HSV-1 DNA with known DNA copy number. ICP0 and VP16 HSV-1 genes were quantified using specific primers. Error bars in B are standard deviations in DNA copy numbers from three independent qPCR reactions repeated at the same conditions. Error bars in C and D are progressed standard deviations.

DOI: https://doi.org/10.7554/eLife.47212.009

that essentially all of HSV-1 capsids bound to NPCs eject their DNA into reconstituted host nuclei, sets the stage for a definitive test of hypothesis for a pressure-driven mechanism of intranuclear viral genome release.

Isolated nuclei in a cytosol solution supplemented with an ATP-regeneration system were incubated with HSV-1 C-capsids with ~18 atm osmotic pressure in the extranuclear solution (generated by 30% w/w PEG 8 kDa). The pull-down assay described above was used to separate viral DNA *fractions a,b, c,* and *d* after incubation for 40 min at 37°C. qPCR was used to quantify the HSV-1 DNA

copy number in each fraction using VP16/UL48 and ICP0 genes. [Note that phenol-chloroform extraction of each viral DNA fraction prior to qPCR analysis removes PEG from DNA samples to avoid any interference from PEG during PCR amplification.] *Figure 5B and D* show that when the capsid pressure is turned off at 18 atm of external osmotic pressure, the ejection of DNA from capsids bound to nuclei is completely suppressed (*fraction d/fractions c+d* ~ 0.2%). The positions of DNA primers were selected to cover most of the HSV-1 genome length and included both S and L regions corresponding to one copy of VP16 (103,163–104,635 bp) and the two copies of ICP0 [copy 1: (2,113 bp-5,388 bp) and copy 2: (120,207 bp-123,482 bp)]. DNA ejection from HSV-1 capsid follows directionality starting at the 151 kb S-end (*Newcomb et al., 2009*). This primer selection ensured that both complete and partial ejection of the HSV-1 genome (151 kb total length) into the nucleus could be detected.

As a control, *Figure 5C* shows that the total DNA copy number added from all four fractions (*a,b, c,d*) separated with the pull-down assay remains the same with and without 30% w/w PEG added to the capsid-nuclei sample. This confirms that no DNA is lost during the DNA fractionation steps due to PEG addition, and therefore the observed reduction in DNA amount in *fraction d* (ejected intranuclear DNA) is entirely attributed to the suppression of DNA ejection from nuclei-bound capsids. As described above, fluorescent microscopy imaging in *Figure 3B* showed that the addition of 30% PEG 8 kDa does not interfere with capsid binding to nuclei. By determining the amounts of DNA injected into nuclei (*fraction d*) and retained in the capsids bound to nuclei (*fraction c*), *Figure 5B* shows that the number of capsids bound to nuclei is in fact slightly increased (*fractions c and d* combined) with 30% PEG addition (the number of unbound capsids in *fraction b* have correspondingly decreased). Enhanced capsid binding to NPCs at nuclei can be explained by the crowding effect induced by PEG molecules, which has been observed to enhance macromolecular binding (*Minton, 2006*). *Fraction a* in *Figure 5B*, corresponding to free viral DNA in the extranuclear solution from broken capsids, is also reduced by PEG addition. This is primarily attributed to the decreased number of unbound capsids in the extranuclear solution and also to increased capsid stability induced by PEG. However, it should also be noted that even without PEG, *fraction a* only constitutes <0.2% of the total viral DNA amount and is at the level of qPCR background noise.

For a final visual demonstration of the osmotic suppression of DNA ejection from HSV-1 capsids into reconstituted cell nuclei, we used ultrathin-sectioning EM. As a negative control, capsids were first incubated with isolated nuclei with added cytosol at 4°C for 40 min without ATP regeneration system. As was previously observed (*Ojala et al., 2000*), these conditions prevented DNA ejection from viral capsids with ~95% of HSV-1 capsids retaining their genomes (*Figure 6*). Next, reproducing the optimized capsid-nuclei binding conditions from the pull-down assay above, purified DNA-filled C-capsids were incubated with isolated rat liver nuclei in the presence of cell cytosol supplemented with ATP regeneration system for 40 min at 37°C. After this incubation, *Figure 6* shows EM micrographs of capsids attached to nuclei, where ~ 62% of capsids are empty with fully ejected DNA when no PEG is present (at least 100 nuclei bound capsids were counted for each sample analysis). The failure to eject DNA from the remaining ~38% of capsids can be attributed to the capsid damage and failure to attach to the NPCs. Indeed, *Figure 6* shows capsids that bind to the nuclear membrane in multilayer clusters, where only the first layer, closest to the nuclear membrane, can dock to the NPCs and eject the DNA. Capsids in the outer layers therefore retain their genomes. By contrast, *Figure 6* shows that when capsids were incubated with reconstituted nuclei (in cytosol supplemented with ATP-regeneration system) for 40 min at 37°C with 30% w/w PEG 8 kDa present, the majority of capsids (~81%) did not eject their genome and retained DNA in the capsid. [In the pull-down assay, the estimated fractions of capsids that ejected DNA without PEG or retained DNA with PEG present, were even higher. This can be attributed to the fact that in the pull-down assay, only capsids that were directly bound to nuclear NPCs were accounted for (*fractions c and d*), separated with several centrifugation steps and multiple washes of nuclei with bound capsids, unlike in the EM analysis where unbound capsids are also present. Qualitatively, however, the EM data supports all of the pull-down assay observations above.]

The NLS transport experiment above (*Section 2*) and the nucleocytoplasmic transport measurements under hyperosmotic stress reported in *Finan et al. (2011)* showed that NPCs' transport functionality is not disrupted by 18 atm PEG-generated osmotic pressure. Here, we further demonstrate that the observed suppression of DNA ejection from capsids into nuclei is caused by the osmotic pressure gradient across the capsid wall, which turns the capsid pressure off, as opposed to PEG

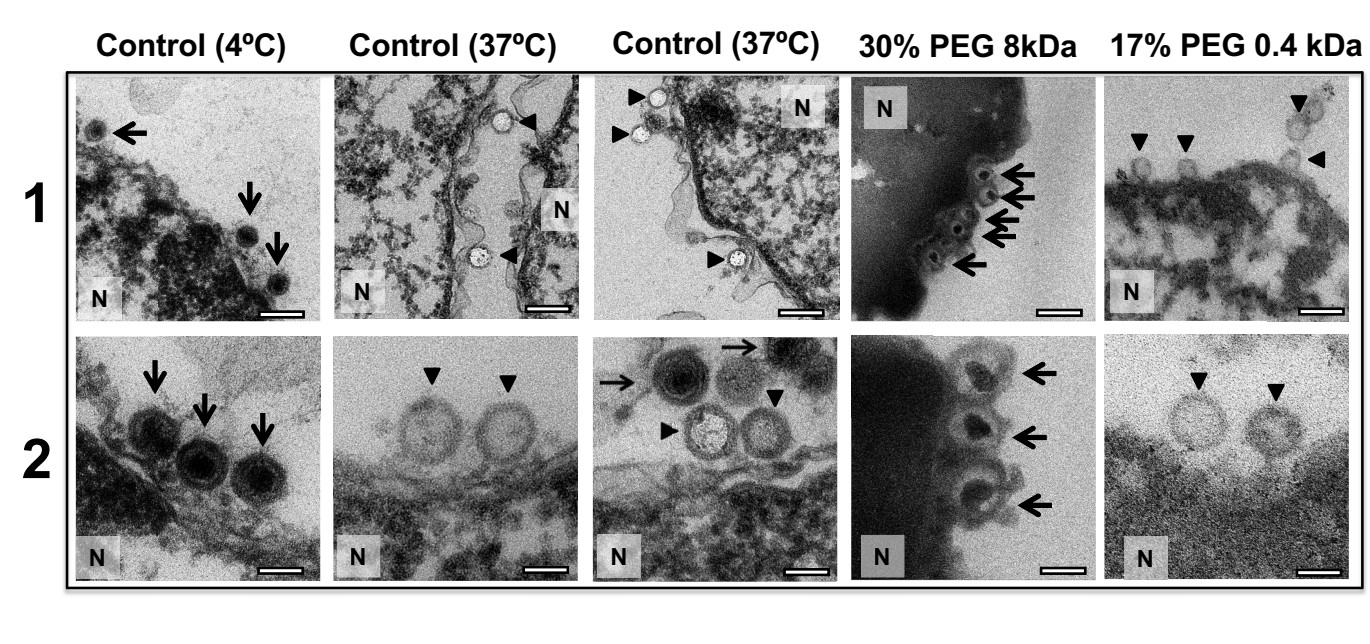

| | Capsids bound to NPC | |
|---|---|---|
| | DNA-filled capsids | Empty capsids |
| Control 4°C | 95 (95%) | 5 (5%) |
| Control 37°C | 38 (38%) | 62 (62%) |
| PEG 8kDa | 81 (81%) | 19 (19%) |
| PEG 0.4 kDa | 40 (40%) | 60 (60%) |

**Figure 6.** Ultrathin-sectioning EM visualization of complete osmotic suppression of DNA ejection from HSV-1 capsids into reconstituted cell nuclei when capsid pressure is 'turned off' by 18 atm osmotic pressure generated by PEG 8 kDa. Negative control at 4°C without added PEG and without ATP-regenerating system, shows that no ejection from nuclei bound C-capsids occurs. Positive control at 37°C shows complete DNA ejection from C-capsids bound to isolated cell nuclei supplemented with cytosol and ATP-regenerating system. EM images show that capsids can bind to the nuclear membrane as individual capsids or in multilayer clusters. Consequentially, only capsids in the first layer that are bound to the NPCs are able to eject their DNA. EM shows that the addition of 30% PEG 8 kDa to reconstituted capsid-nuclei system inhibits DNA ejection from HSV-1 C-capsids into host nuclei through the NPC. In all samples, capsids and nuclei were incubated for 40 min. Thin arrows show DNA-filled capsids, and bold arrows show empty capsids that ejected DNA. 1. Bar 500 nm. 2. Bar 90 nm. Representative EM images are shown. At least 100 capsids bound to NPCs were counted for each sample's statistical analysis, shown in the table below.

DOI: https://doi.org/10.7554/eLife.47212.010

The following figure supplement is available for figure 6:

**Figure supplement 1.** Calculated DNA pressures in herpesvirus capsids.

DOI: https://doi.org/10.7554/eLife.47212.011

itself and/or its osmotic pressure effect blocking the NPC channel and interfering with the transport functionality. To show this, we repeated the capsid-nuclei binding experiment above but this time PEG 8 kDa was replaced with PEG 400 Da. At 17% w/w, PEG 400 generates 18 atm of osmotic pressure (see https://brocku.ca/researchers/peter_rand/). This osmotic pressure was required for

complete suppression of DNA ejection with PEG 8 kDa. However, as mentioned above, HSV-1 capsid pore size has a MW cutoff of ~4000 Da (*Trus et al., 1996*; *Heymann et al., 2003*); therefore, PEG 400 Da permeates the capsid (unlike PEG 8 kDa). Accordingly, even when capsids bound to reconstituted nuclei are incubated at 18 atm osmotic pressure with PEG 400 Da, there will be no osmotic pressure gradient (between inside and outside the capsid) needed to cancel the DNA pressure in the capsid (*Evilevitch et al., 2008*). Indeed, when isolated nuclei reconstituted with cytosol and an ATP-regeneration system were incubated with C-capsids for 40 min at 37°C with 17% w/w PEG 400 Da added to the solution, ultrathin-sectioning EM (*Figure 6*) showed that, despite18 atm osmotic pressure surrounding the capsids,~60% of capsids were empty (ejected their DNA). This is equivalent to the fraction of empty capsids that ejected DNA after incubation of capsids with reconstituted nuclei for 40 min at 37°C without PEG addition (~62% empty capsids). This observation further validates the assumption that it is the osmotic pressure gradient that suppresses DNA ejection through the NPCs by 'turning off' the capsid pressure and not the interference with NPC transport functionality.

Combined, the pull-down assay and the EM data clearly demonstrate that viral DNA ejection into host nuclei from HSV-1 capsids bound to NPCs is completely blocked when the intracapsid genome pressure is turned off through an osmolyte addition. This proves that viral DNA translocation from herpesvirus capsids across the nuclear membrane is driven by intracapsid pressure.

## Conclusions

All eukaryotic DNA viruses, with the exception of poxviruses, deliver and replicate their genomes within the nucleus (*Hennig and O'Hare, 2015*). In addition, retro-viruses transport their genomes across nuclear membranes in order to replicate after they have reversely transcribed their single-stranded (ss) RNA to dsDNA. The mechanism of this most significant step of infection, viral DNA entry into the host nucleus, remains poorly understood for the majority of viruses (*Hennig and O'Hare, 2015*). The reconstituted capsid-nuclei experiments above provide a demonstration of a nuclear entry mechanism of DNA from HSV-1, driven by high mechanical pressure of the encapsidated viral genome. Despite previous measurements of intracapsid DNA pressure in several types of viruses (*Evilevitch et al., 2003*; *Bauer et al., 2013*), the role that capsid pressure plays for intranuclear viral genome delivery has not been demonstrated until now.

The aim of this work was to demonstrate that capsid pressure is critical for initiation of DNA ejection from a herpesvirus capsid into a host nucleus. However, other factors may contribute to complete internalization of the viral genome into the nucleus once the first portion of herpes DNA is released through the NPC channel by the capsid pressure. As shown in *Figure 2*, capsid pressure is rapidly reduced with the increasing fraction of ejected DNA. At osmotic pressures of 3–4 atm, equivalent to that of the cellular cytoplasm surrounding the capsid (*Jeembaeva et al., 2008*), only ~50% of DNA is ejected. This leaves the question as to how the rest of the genome is released. [However, not all macromolecules generating the osmotic pressure in the cell are large enough not to penetrate the capsid, which, as discussed above, is required for osmotic suppression of DNA ejection. Thus, we anticipate that a DNA fraction larger than 50% is ejected by DNA pressure into the crowded cellular environment]. We had previously found that the crowded cellular environment (which contributes to the osmotic pressure [*Parsegian et al., 2000*]), combined with the presence of DNA-binding proteins in the cell, lead to instant condensation of the incoming viral DNA. This DNA condensation exerts a significant pulling force on the rest of the DNA, facilitating its complete internalization in the cell (*Jeembaeva et al., 2008*). Further, a recent theoretical study proposed that DNA ejection could be described by a two-step process, where the first portion of DNA is ejected by DNA-DNA repulsive pressure, followed by a slower process of anomalous diffusion of condensed viral genome in the crowded cytoplasm (*Chen et al., 2018*). These effects can be investigated in the future using the reconstituted nucleus system.

High intracapsid DNA packing density resulting in tens of atmospheres of pressure is a distinctive trait of all nine human herpesviruses. In *Figure 6—figure supplement 1*, we calculated capsid DNA pressures in several types of herpesviruses using analytical expressions in refs. (*Tzlil et al., 2003*; *Purohit et al., 2003*) and EM measured values for inner capsid diameters (*Booy et al., 1991*; *Germi et al., 2012*; *Yi et al., 2017*) to compute DNA-DNA electrostatic repulsive force and bending stress. As a reference pressure, the calculated HSV-1 DNA pressure is in agreement with our measured value of 19 atm (*Bauer et al., 2013*). The differences in computed pressures (ranging

from ~14 atm for VZV to ~90 atm for EBV) are related to the variation in DNA packing density of these viruses (*Booy et al., 1991*; *Germi et al., 2012*; *Yi et al., 2017*). This strongly suggests that pressure-driven entry of viral DNA into the host nucleus during infection is universal to all herpesviruses.

Other types of viruses also involve replication steps dependent on the pressurized state of the intracapsid genome. For instance, during genome packaging, reoviruses replicate ssRNA to dsRNA inside the capsid, which results in genome packaging densities similar to that of herpesviruses (*Prasad et al., 1996*). Such intracapsid replication could be regulated, at least in part, by generation of internal pressure resulting from the increasing genome packaging density as newly synthesized dsRNA continues to fill the internal capsid volume. Another example is HIV, where similar to herpesviruses, HIV capsids dock to NPCs at the nucleus and release transcribed dsDNA through the NPC channel (*Rankovic et al., 2017*). It was recently shown that the reverse transcription process from ssRNA to dsDNA inside the HIV capsid is associated with increasing internal DNA pressure (*Rankovic et al., 2017*). By demonstrating the central function of herpes capsid pressure for intranuclear viral DNA entry, combined with assays developed in this work, we provide a platform for analysis of pressure-regulated replication in many viruses that afflict humans and animals.

## Materials and methods

### Cells and viruses

African green monkey kidney cells (Vero; ATCC CCL-81 from American Type Culture Collection, Rockville, MD) and BHK-21 cells (ATCC CCL-10; from American Type Culture Collection, Rockville, MD) were cultured at 37°C in 5% CO2 in Dulbecco's modified Eagle's medium (DMEM; Life Technologies) supplemented with 10% fetal bovine serum (FBS; Gibco), 2 mM L-glutamine (Life Technologies), and antibiotics (100 U/ml penicillin and 100 μg/ml streptomycin; Life Technologies). The KOS strain of HSV-1 was used as the wild-type strain. The K26GFP HSV-1 recombinant virus (gift from Dr. Fred Homa, University of Pittsburgh), that carries a GFP tag on the capsid protein VP16 was used in fluorescence studies. All viruses were amplified on Vero cells, and titers were determined on Vero cells by plaque assay. Viral plaque assays were carried out as follows: Viral stocks were serially diluted in DMEM. Aliquots were plated on 6-well trays of Vero cells for 1 hr at 37°C. The inoculum was then replaced with 40% (v/v) carboxymethylcellulose in DMEM media. HSV-1 plaque assays were incubated for 3–4 days. The monolayers were stained for 1 hr with crystal violet stain (Sigma-Aldrich). After removal of the stain, the trays were rinsed with water and dried, and plaques were counted.

### HSV-1 C-capsid isolation

Purification of HSV-1 capsids was previously described (*Bauer et al., 2013*). African green monkey kidney cells (Vero) were infected with either HSV-1 KOS strain or a K26GFP HSV-1 recombinant virus at a multiplicity of infection (MOI) of 5 PFU/cell for 20 hr at 37°C. Cells were scraped into solution and centrifuged at 3500 r.p.m. for 10 min in a JLA-16.250 rotor. The cell pellet was re-suspended in 20 mM Tris buffer (pH 7.5) on ice for 20 min and lysed by addition of 1.25% (v/v) Triton X-100 (Alfa Aesar) for 10 min on ice. Lysed cells were centrifuged at 2000 rpm for 10 min and the nuclei pellets were re-suspended with 1x protease inhibitor cocktail (Complete; Roche) added. Nuclei were disrupted by sonication for 30 s followed by treatment with DNase I (Thermo-Fisher) for 30 min at room temperature. Large debris were cleared by brief centrifugation, and the supernatant was spun in a 20–50% (w/w) sucrose gradient in TNE buffer (500 mM NaCl, 10 mM Tris, 1 mM Na$_2$EDTA, pH 8.0) at 24,000 rpm in a SW41 rotor for 1 hr. The C-capsid band was isolated by side puncture, diluted in TNE buffer and centrifuged at 23,000 rpm for an additional 1 hr. Capsids were re-suspended in a preferred capsid binding buffer (CBB: 20 mM HEPES-KOH with pH of 7.3, 80 mM K-acetate, 2 mM DTT, 1 mM EGTA, 2 mM Mg-acetate, 1 mM PMSF, and 1X CLAP cocktail).

### Osmotic suppression of DNA ejection and PFGE analysis

HSV-1 C-capsids along with varying concentrations of 8 kDa MW polyethylene glycol (PEG) (Fisher) were incubated at 37°C for 1.5 hr with trypsin and DNase I as previously described (*Bauer et al., 2013*). The corresponding osmotic pressure (Π) as a function of the PEG w/w percentage ($w$) was

determined by the empirical relation (*Evilevitch et al., 2003*) $\Pi(atm) = -1.29\ G^2 T + 140\ G^2 + 4G$, where $G = w/(100 - w)$ and $T$ is the temperature (°C). Non-ejected DNA was extracted from capsids by addition of 10 mM ethylenediaminetetra-acetic (EDTA) (Duchefa), 0.5% (*w/v*) SDS (Sigma), and 50 µg/mL protease K (Amresco) followed by a 1 hr incubation at 65°C. The length of osmotically suppressed DNA within capsids was determined by pulse field gel electrophoresis using a Bio-Rad CHEF II DR at 6 V/cm with initial and final switch times of 4 and 13 s respectively. Gels were stained with SybrGold and size estimations performed with UVP VisionWorksLS software using the Mid-range molecular weight standard from New England BioLabs as a reference.

## Rat liver nuclei isolation and cytosol preparation

Nuclei from rat liver were isolated as adapted from previously described protocol (*Ojala et al., 2000*). The intactness of nuclei was confirmed by light microscopy, EM (electron microscopy) and FM (fluorescence microscopy) by staining the nuclei with DAPI and by their ability to exclude fluorescently tagged (Fluorescein isothiocyanate) 70 kDa dextran. The cytosol was separately prepared using BHK-21 cells.

## Reconstituted capsid-nuclei system

An in-vitro viral HSV-1 DNA translocation system was built in which HSV-1 genome was released into nucleoplasm in a homogenate solution mimicking cytoplasm environment, see details in previously described protocol in *Ojala et al. (2000)*. In a typical system, rat liver cell nuclei were incubated C-capsids (HSV-1 or GFP-labeled HSV-1), containing: (i) cytosol, (ii) BSA, (iii) ATP-regeneration system, see details in *Ojala et al. (2000)*. The system was incubated 37°C for 40 min sufficient for capsid binding to nuclei. For inhibition studies, wheat germ agglutinin (WGA) was pre-incubated with the nuclei prior to addition of C-capsids.

## NPC transport functionality

We verified that NPC transport functionality was not disrupted by 18 atmospheres of osmotic pressure generated by PEG. We performed an in-vitro import assay to evaluate the nuclear import activity of NPCs using the nuclear localization signal (NLS) (*Miyamoto et al., 2002*). Purified rat liver nuclei were incubated with cytosolic extracts (as a source of soluble import factors) supplemented with ATP-regeneration system and a purified GST-NLS-EGFP recombinant protein, which contains the nuclear localization signal (NLS) of the simian virus 40 T antigen fused with glutathione S-transferase (GST) and EGFP. This protein was used as a positive import substrate, since it is transported into the nucleus by an active non-diffusion mechanism and can be detected by fluorescence microscopy (*Miyamoto et al., 2002*; *Tsuji et al., 2007*; *Vázquez-Iglesias et al., 2009*).

## Fluorescence microscopy

For fluorescence imaging of the reconstituted capsid-nuclei system, GFP-labeled HSV-1 C-capsids were used. After incubation of capsids with the nuclei as described above, the buffer system containing purified GFP-labeled C-capsids and nuclei were loaded onto cover-slips (Mab-Tek). The nuclei were stained with DAPI for 5 min before imaging. Overlay of the confocal 488 (for GFP emitted signal) and 358 (for DAPI emitted signal) channels show the localization of viral capsids onto the nucleus. Images were captured with a Nikon A1R laser-scanning confocal microscope. For inhibition studies with wheat germ agglutinin (WGA), the nuclei were pre-incubated with 0.5 mg of WGA/ml for 20 min on ice before addition of GFP-labeled HSV-1 C-capsids.

## Super Resolution-Structured illumination microscopy (SR-SIM)

After incubation of nuclei with GFP-labeled C-capsids, the complete binding mixture was loaded onto chamber slides (Mab-Tek) and the samples were immediately imaged for GFP and DAPI by using 405 nm and 488 nm excitation wavelengths with a Zeiss Elyra S1 microscope with a 64X-oil immersion lens. The images were captured on a sCMOS PCO Edge camera. The images were processed using the Structured Illumination module of the Zeiss (software Zen ver. 2011) software to obtain the super-resolved images of GFP-capsids bound to nuclei. The spatial resolution of the instrument is 120 nm. To generate 3D reconstructions, image stacks (1 µm) were acquired in Frame Fast mode with a z-step of 110 nm and 120 raw images per plane. Raw data was then

computationally reconstructed using the Zen software to obtain a super-resolution 3D image stack. The Fiji-ImageJ software was used to generate the histogram of the cross-section profile for the GFP-labeled C-capsid signal.

## Electron microscopy (EM)

After binding of capsids to nuclei, the samples were washed with CBB buffer. The supernatant was then removed and replaced with fixative (2.5% EM-grade glutaraldehyde and 2.0% EM-grade formaldehyde in 0.1 M sodium cacodylate buffer, pH 7.4) for 3 hr at 4°C. The fixative was then removed and replaced with 1% osmium tetroxide in buffer for 90 min. Each sample was then subjected to 10 min buffer rinse, after which it was placed in 1% aqueous uranyl acetate and left overnight. The next day, each sample was dehydrated by using a graded ethanol series and propylenoxid. The nuclear pellets were embedded in Epon prior to cutting. Ultrathin Epon sections on grids were stained with 1% aqueous uranyl acetate and lead citrate (*Reynolds, 1963*). After the grids dried, areas of interest were imaged at 120 kV, spot three using a Tietz 2k × 2 k camera mounted on a Philips/FEI (now Thermo Fisher FEI) CM200 transmission electron microscope.

## Capsid pull-down assay

After capsids-nuclei incubation, as described above, the system was centrifuged at 3,000 rpm to spin down the nuclei-associated capsids, and the supernatant was collected separately as the extranuclear solution. Nuclei pellet was washed extensively in CBB buffer at 4°C to remove excessive osmolytes in the pellet (all the steps were carried at 4°C to minimize DNA ejections after the incubation stage). The pellet was then re-suspended and incubated for 20 min. in 1x reticulocyte standard buffer (RSB: 10 mM Tris of pH 7.5, 10 mM KCl, 1.5 mM MgCl2, 0.5% NP-40 substitute) to lyse the nuclear membrane. Both extranuclear supernatant solution and lysed nuclear pellet were then incubated with 5 µL of an anti-HSV1/2 ICP5/UL19 antibody overnight at 4°C. The next day, 50 µL of 50% Protein A bead slurry (Sigma-Aldrich) was added to each sample to capture viral capsid-antibody complex. Protein A bead complexes were then centrifugated (1500 rpm, 5 min), and the supernatants were collected (sample b and d in *Figure 2*). In parallel, the pelleted beads were re-suspended in Proteinase K (Amresco) solution to digest the capsid and let viral DNA diffuse into the solution (Fractions a and c). Then, DNA from each sample was recovered by phenol–chloroform extraction, precipitated with ethanol and re-suspended in DNase-free ultrapure water.

This in-vitro assay successfully divides the HSV-1 DNA into four fractions: (a) DNA extracted from capsids that failed to bind to NPCs, (b) free DNA in the extranuclear solution from broken capsids, (c) DNA retained inside the capsids that were bound to nuclei but did not eject DNA, and (d) DNA ejected from capsids into the nucleoplasm. Extracted viral DNAs from Fractions a, b, c, d were quantified by qPCR analysis for DNA level by custom TaqMan assays. Viral genes VP16 and ICP0 were quantified with specific primers (gift from Bernard Roizman lab). The assays were performed by using a StepOnePlus system (Applied Biosystems) and were analyzed with a software provided by the supplier. A WT HSV-1 DNA with known viral copy number was used to generate a standard curve and to calculate the viral gene copy number of the unknown samples.

## Acknowledgements

We would like to thank Bernard Roizman for help with designing and conducting the pull-down assay experiments. We greatly acknowledge Fred Homa, Jamie Huffman, Roger Lippé, Päivi Ojala, Lindsay Smith, and Beate Sodeik, for sharing their viral strains, protocols, and important advice in preparation of this manuscript. We thank Scott Robinson for help with ultra-thin sectioning EM imaging. EM imaging was done at the Beckman Institute core facility at UIUC. We thank Glenn Fried and Austin Cyphersmith for help with SR-SIM imaging at the Carl R Woese Institute for Genomic Biology. Krista Freeman, Dong Li and Dave Bauer are acknowledged for help with the initial optimization of experimental conditions. Adrián González Rodríguez is acknowledged for help with the calculation of pressures in herpesviruses. Funding for this work was provided by the National Science Foundation CHE-1744061 (to AE) and Swedish Research Council grants (VR) 621-2014-5537 and 349- 2014–3962 (to AE).

## Additional information

### Funding

| Funder | Grant reference number | Author |
|---|---|---|
| National Science Foundation | CHE-1744061 | Alex Evilevitch |
| Vetenskapsrådet | 621-2014-5537 | Alex Evilevitch |
| Vetenskapsrådet | 349- 2014-3962 | Alex Evilevitch |

The funders had no role in study design, data collection and interpretation, or the decision to submit the work for publication.

### Author contributions

Alberto Brandariz-Nuñez, Formal analysis, Investigation, Visualization, Methodology, Writing—original draft; Ting Liu, Te Du, Formal analysis, Investigation, Methodology; Alex Evilevitch, Conceptualization, Resources, Data curation, Formal analysis, Supervision, Funding acquisition, Validation, Investigation, Visualization, Methodology, Writing—original draft, Project administration, Writing—review and editing

### Author ORCIDs

Alex Evilevitch 🆔 https://orcid.org/0000-0002-0245-9574

### Decision letter and Author response

Decision letter https://doi.org/10.7554/eLife.47212.015
Author response https://doi.org/10.7554/eLife.47212.016

## Additional files

### Supplementary files

• Supplementary file 1. qPCR data for DNA copy numbers in each fraction obtained with the pull-down assay. *The efficiency of pull-down assay was separately assessed by DNase protection method. This was shown by adding DNase I to Fraction b after immunoprecipitation step. DNase I digests all free DNA from broken capsids, but DNA inside the capsids, which were not immunoprecipitated, remains intact (DNase does not permeate the capsid shell). Protease K and phenol-chloroform extraction release the encapsidated DNA, which is then quantified with qPCR and reflects the number of capsids that were not pulled-down. This fraction constitutes only 1–6% of the total HSV-1 DNA amount in Fraction b prior to immunoprecipitation. Standard deviations in DNA copy numbers were obtained from three independent qPCR reactions repeated at the same conditions for each sample.
DOI: https://doi.org/10.7554/eLife.47212.012

• Transparent reporting form
DOI: https://doi.org/10.7554/eLife.47212.013

### Data availability

All data is provided within the text and supplemental material.

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
