## [Decision Letter]

Thank you for submitting your article "Pressure-Driven Release of viral genome into a host nucleus-mechanism of herpes infection" for consideration by *eLife*. Your article has been reviewed by Arup Chakraborty as the Senior Editor, a Reviewing Editor, and three reviewers.

The reviewers have discussed the reviews with one another and the Reviewing Editor has drafted this decision to help you prepare a revised submission.

The reviewers are in agreement that the experiments reported here have been well-designed and carried out, and that they shed light on the important process of pressure-driven viral genome delivery. In particular they demonstrate genome translocation from envelope-free HSV1 capsids through nuclear pore complexes (NPCs) into nuclei isolated from rat liver cells supplemented with importin-β and ATP regeneration components necessary for NPC binding and opening. After establishing delivery of genomic DNA to the nuclei, this delivery is suppressed by adding a critical PEG concentration (established in separate in vitro work with purified capsids and trypsin for triggering genome release).

The reviewers recommend publication subject to the authors responding to the several following concerns and suggestions:

1) The authors should drop all mention of "infection" in the account of their work, because of their system being restricted – by design, as appropriate – to an in vitro reconstitution of DNA delivery to isolated nuclei.

2) Similarly, their claims that the work is relevant to new anti-viral therapies involving polyvalent ions (or by any other means, for that matter) should be dropped.

3) The Van Valen at al. (2012) and Chen et al. (2018) papers should be cited and their relevance to the present work described briefly, particularly with regard to their discussion of pressure-driven genome delivery in a full biological context and their calling attention to other physical mechanisms – in addition to pressure driving the initial stages – for DNA translocation.

4) In this context, what do the present experiments tell us about the length of genome delivered?

5) The authors should discuss alternative explanations of the effect of PEG in their experiments, such as osmotic stress of the nuclei themselves, or interfering with the mechanism of NPC activity in translocating DNA.

6) What is the effect of osmotic pressure in the nuclei themselves, on the quantitation of suppression reported in the present work and on the length of genome delivered?

7) Language such as "important discovery" and "first demonstration" should be avoided.

---

## [Author Response]

The reviewers recommend publication subject to the authors responding to the several following concerns and suggestions:1) The authors should drop all mention of "infection" in the account of their work, because of their system being restricted – by design, as appropriate – to an in vitro reconstitution of DNA delivery to isolated nuclei.

The reconstituted nucleus system accurately reproduces capsids-nuclei binding and nuclear transport of the herpes genome into living cells. Ojala et al. (2000) finds good agreement in experimental comparison between the reconstituted nuclei system (which includes all cellular components required to facilitate HSV-1 capsids’ specific binding to NPCs, opening of the capsid portal, and translocation of viral DNA through the NPC pore) and HSV-1 binding to NPCs followed by DNA release in living cells. Therefore, our study of the DNA ejection step from HSV-1 into reconstituted cell nuclei reproduces the herpes DNA ejection step in living cells.

Indeed, we show here that the DNA pressure in HSV-1 capsids powers ejection of the viral genome into a reconstituted cell nucleus. Thus, we have only isolated one central step of the herpes infectious cycle – viral DNA translocation. However, the term “infection” denotes introduction of nucleic acid into a host cell by a virus (not necessarily with genome replication). In fact, primary infection by several types of herpesviruses (including HSV-1) is latent, i.e. the herpes genome is translocated into the host nucleus without subsequent genome replication (Adam, Marr and Gerace, 1990). Therefore, the osmotic suppression assay combined with the reconstituted nucleus system present a platform for the analysis of a pressure-dependent mechanism of herpesvirus’ infectious cycle focused on the DNA translocation step into the host. This has now also been explained in the text.

Following the editors’ recommendations, we have removed the word “infection” at several places in the text and have replaced it with “leading to infection”. We have also clarified in the text what denotes the term “infection”, specifically with regard to latent infection in herpesviruses. We hope that this revised presentation is acceptable and provides an accurate description of our system.

2) Similarly, their claims that the work is relevant to new anti-viral therapies involving polyvalent ions (or by any other means, for that matter) should be dropped.

We have now removed all mention of this throughout the text.

3) The Van Valen at al. (2012) and Chen et al. (2018) papers should be cited and their relevance to the present work described briefly, particularly with regard to their discussion of pressure-driven genome delivery in a full biological context and their calling attention to other physical mechanisms – in addition to pressure driving the initial stages – for DNA translocation.

The findings in their Van Valen et al. (2012) and Chen et al. (2018) papers have now been described and cited in both the Introduction and Conclusions sections in the manuscript.

However, the focus of this manuscript is to demonstrate that capsid pressure is critical for viral DNA delivery into a cell nucleus, which has not been shown experimentally before. We show this by “turning off” the pressure in the capsid and observing that no intranuclear DNA ejection occurs. Indeed, as discussed in Chen et al. (2018), capsid pressure might only be responsible for initial stages of DNA delivery, where the rest of the viral genome is internalized in the nucleus by other processes, such as anomalous diffusion of the condensed phage genome into the crowded bacterial cytoplasm. As mentioned above, the focus of our manuscript was not to explain the mechanism of translocation of the entire viral genome but only to show the role of capsid pressure. This has now been further clarified in the Conclusions section of this manuscript.

4) In this context, what do the present experiments tell us about the length of genome delivered?

We have now clarified in the text that:

“The positions of DNA primers were selected to cover most of the HSV-1 genome length and included both S and L regions corresponding to one copy of VP16 (103,163-104,635 bp) and the two copies of ICP0 [copy 1: (2,113 bp-5,388 bp) and copy 2: (120,207 bp-123,482 bp)]. DNA ejection from HSV-1 capsid follows directionality starting at the 151 kb S-end (Newcomb et al., 2009). This primer selection ensured that both complete and partial ejection of the HSV-1 genome (151 kb total length) into the nucleus could be detected.”

5) The authors should discuss alternative explanations of the effect of PEG in their experiments, such as osmotic stress of the nuclei themselves, or interfering with the mechanism of NPC activity in translocating DNA.

We have further clarified this in subsection “Reconstituted capsid-nuclei system” by adding the explanation below along with one new experiment:

Finan, Leddy and Guilak (2011) demonstrates that nuclear transport through the NPCs is not negatively affected by hyper-osmotic conditions corresponding to those used in our study (~20 atm). Specifically, the authors reported that, under hyper-osmotic stress, the nuclear size decreased while nuclear lacunarity increased, indicating expansion in the pores and channels interdigitating the chromatin. As a result, the rate of nucleocytoplasmic transport increased but only due to the change in the nucleus geometry, providing a shorter effective diffusion distance. This sensitivity to hyperosmotic conditions concerned both passive and active transport across the NPCs. At the same time, the authors found that diffusivity within the nucleus was insensitive to the osmotic environment. In agreement with these studies (Finan, Leddy and Guilak, 2011), we observed that, under hyperosmotic conditions (~18 atm at 30% w/w PEG 8 kDa), the nuclei slightly shrunk (Figure 3—figure supplement 1A). However, the sub-nuclear structure of heterochromatin DNA was essentially unchanged upon addition of PEG, as visualized by a DAPI stain of nuclear DNA (Figure 3—figure supplement 1A, second row). We also confirmed that the integrity of the nuclei was not affected by the addition of 30% w/w PEG 8kDa by showing that fluorescently labeled 70 kDa dextran is excluded from the nuclei interior with nuclei remaining intact and structured, see Figure 3—figure supplement 1B. Finally, we showed that the full transport functionality of NPCs is maintained in the reconstituted nuclei system at an osmotic pressure of ~18 atm generated by PEG. This was verified with a fluorescently labeled nuclear localization signal (NLS) (data not shown) (Sekine et al., 2017). Purified GST-NLS-EGFP recombinant protein, which contains the NLS of the simian virus 40 T antigen fused with glutathione S-transferase (GST) and EGFP fluorescent protein, was used. Purified rat liver nuclei were incubated with cytosolic extracts (as a source of soluble import factors) supplemented with an ATP-regeneration system and a purified GST-NLS-EGFP recombinant protein at ~18 atm external osmotic pressure generated by PEG. This protein was used as a positive import substrate. It was fully transported into the nucleus through the NPC by an active mechanism, which was detected by fluorescence microscopy (Finlay et al., 1987; Finan, Leddy and Guilak, 2011; Miyamoto et al., 2002) (see details in the Materials and methods section).

The NLS transport experiment (subsection “Reconstituted capsid-nuclei system”) and the nucleocytoplasmic transport measurements under hyperosmotic stress reported in Finan, Leddy and Guilak (2011) showed that NPCs’ transport functionality is not disrupted by 18 atm PEG-generated osmotic pressure. Here, we further demonstrate that the observed suppression of DNA ejection from capsids into nuclei is caused by the osmotic pressure gradient across the capsid wall, which turns the capsid pressure off, as opposed to PEG itself and/or its osmotic pressure effect blocking the NPC channel and interfering with the transport functionality. To show this, we repeated the capsid-nuclei binding experiment but this time PEG 8 kDa was replaced with PEG 400 Da. At 17% w/w, PEG 400 generates 18 atm of osmotic pressure (see https://brocku.ca/researchers/peter_rand/). This osmotic pressure was required for complete suppression of DNA ejection with PEG 8 kDa. However, as mentioned above, HSV-1 capsid pore size has a MW cutoff of ~4000 Da (Trus et al., 1996; Heymann et al., 2003); therefore, PEG 400 Da permeates the capsid (unlike PEG 8 kDa). Thus, even when capsids bound to reconstituted nuclei are incubated at 18 atm osmotic pressure with PEG 400 Da, there is no osmotic pressure gradient between the interior and the exterior of the capsid, which needed to generate the osmotic gradient required to cancel the DNA pressure in the capsid (Evilevitch et al., 2008). Thus, isolated nuclei reconstituted with cytosol and an ATP-regeneration system were incubated with C-capsids for 40 min at 37°C with 17% w/w PEG 400 Da added to solution. Ultrathin-sectioning EM in Figure 6 shows that, despite 18 atm osmotic pressure surrounding the capsids, ~60% of capsids were empty (ejected their DNA), equivalent to the fraction of empty capsids that ejected DNA after incubation of capsids with reconstituted nuclei for 40 min at 37°C without PEG addition (~62% empty capsids). This observation further validates the assumption that it is the osmotic pressure gradient that suppresses DNA ejection through the NPCs by “turning off” the capsid pressure and not the interference with NPC transport functionality.

6) What is the effect of osmotic pressure in the nuclei themselves, on the quantitation of suppression reported in the present work and on the length of genome delivered?

The effect of osmotic pressure on nuclei has been further clarified in the manuscript.

We have also provided clarification in the Conclusions section, addressing the question of the effect of osmotic pressure on the ejected DNA length:

“The aim of this work was to demonstrate that capsid pressure is critical for initiation of DNA ejection from a herpesvirus capsid into a host nucleus. […] These effects can be investigated in the future using the reconstituted nucleus system.”

7) Language such as "important discovery" and "first demonstration" should be avoided.

This language has been removed throughout the text, except at one place in the abstract where we specifically clarify that this is the first demonstration for eukaryotic viruses. We also state “to our knowledge” as suggested by reviewers.